# Activation of Irrigants in Root Canals with Open Apices: A Narrative Review

**DOI:** 10.3390/jcm13216611

**Published:** 2024-11-04

**Authors:** Dina Abdellatif, Massimo Pisano, Luigi Cecere, Valentino Natoli, Davide Mancino, Eduard Euvrard, Alfredo Iandolo

**Affiliations:** 1Service of Maxillofacial Surgery, Stomatology and Hospital Odontology, Laboratoire Sinergies, University of Franche-Comté, 25000 Besançon, France; edouard.euvrad@univ-fcomte.fr (E.E.); alfredo.iandolo@univ-fcomte.fr (A.I.); 2Department of Medicine, Surgery and Dentistry, “Scuola Medica Salernitana”, University of Salerno, 84081 Salerno, Italyluigi.cecere97@gmail.com (L.C.); 3Department of Dentistry, School of Biomedical and Health Sciences, European University of Madrid, 28670 Madrid, Spain; 4Faculty of Dental Surgery, Federation of Medicine Translational of Strasbourg, Federation of Materials and Nanoscience of Alsace, University of Strasbourg, 67084 Strasbourg, France; davidemancino@icloud.com

**Keywords:** NaOCl, endodontics, open apices, irrigant activation

## Abstract

Performing endodontic treatment on a tooth with an immature apex can be challenging due to the risk of irrigant extrusion beyond the apex. This narrative review investigates the over-apex extrusion of activated irrigants in teeth with open and immature apices and aims to provide crucial insights for practice and research. Two hundred fifty-two results were found from the electronic search. Sixteen duplicates were removed before selection, and 24 were excluded. Of the 212 remaining results, the full text was evaluated for eligibility. One hundred ninety-four results were excluded: in one hundred sixty-five, irrigant activation was not performed; twenty-nine were animal studies; in seven, irrigant activation was not performed to disinfect the canal. Finally, 13 studies were included. They cover a range of topics, from the types of irrigants used to the techniques of activation, and their findings contribute to our understanding of the risks and consequences of irrigant extrusion. All the activation techniques investigated can, to varying degrees, cause the irrigant to extrude beyond the apex. Extrusion may not always be clinically relevant; however, the consequences of excessive irrigant leakage from the apex are dangerous, so try to assess all the variables that may cause it and adopt techniques to reduce it.

## 1. Introduction

For long-term success of endodontic treatment, the endodontic disinfection phase is crucial [1]. Endodontic failure is due to the persistence of bacteria resulting from ineffective disinfection [1]. In addition to endodontic treatment, chemical disinfection is also performed in other procedures such as pulpal regeneration and intentional transplantation [2,3].

Performing endodontic treatment on a tooth with an immature apex can be difficult because the diameter of the apex does not allow for complete apex closure [4]. A further complication is the extrusion beyond the apex of the irrigant used to disinfect the canal [5]. The main irrigants used to disinfect the endodontic space are sodium hypochlorite, 17% ethylenediaminetetraacetic acid and 2% chlorhexidine [6]. The concentration of sodium hypochlorite can vary greatly. Higher hypochlorite concentrations (5.25 to 6 percent) allow for greater disinfection but also increase side effects in the case of extrusion over the apex [7]. The use of irrigants, especially if activated, makes it possible to better disinfect the endodontic complex [8] by reducing dentine removal with shaping and avoiding weakening canal walls that are already thin because they are not mature [4]. Techniques to activate irrigants are many and varied: manual activation, heating, negative apical pressure, subsonic techniques, sonic techniques, ultrasonic techniques, and laser techniques [6]. Based on these considerations, this narrative review aims to investigate the over-apex extrusion of activated irrigants in teeth with open apices.

## 2. Materials and Methods

Scientific articles on activating irrigants in tooth roots with open apices from 21 August until 15 September 2023 were meticulously collected using comprehensive search engines such as PubMed, Web of Science, and Scopus. This detailed process ensures the validity of this review.

The keywords used, for all the database, are “Open Apex” OR “Immature Apex” OR “Immature Teeth” AND “Irrigation Activation” OR “Activated Irrigation” OR “Irrigation OR Activation”.

The inclusion criteria are as follows: in vivo, in vitro, and ex vivo studies; studies on humans or human teeth; articles published in indexed scientific journals.

No time filters were applied to the search. Only articles in English were selected.

References were exported and managed using Mendeley Reference Manager software, version 2.120.3, copyright Elsevier Ltd.

Three independent reviewers were responsible for selecting the studies and extracting the data. If they disagreed on any aspect, they resolved it through discussion, and if consensus was not reached, another author was consulted to ensure the integrity and thoroughness of the review process.

## 3. Results

### 3.1. Study Selection

From the electronic search, 252 results were found: 230 from PubMed, 18 from Scopus, and 4 from Web of Science. Sixteen duplicates were removed before selection. Of the remaining 236 titles and abstracts, 24 were excluded. Of the 212 remaining results, the full text was evaluated for eligibility. One hundred ninety-four results were excluded:

In 165, irrigant activation was not being performed.

Twenty-nine were animal studies.

In 7, irrigant activation was not being performed to disinfect the canal.

Eventually, 13 studies [9,10,11,12,13,14,15,16,17,18,19,20,21] met the inclusion criteria.

Figure 1 summarises the flowchart of the studies selection. 

An additional electronic search on the Google Scholar database was performed for the discussion section.

### 3.2. Study Characteristics

Of the thirteen studies [9,10,11,12,13,14,15,16,17,18,19,20,21] in this review, six are in vivo studies [9,10,11,12,13,14] on 10 teeth, four are ex vivo studies [15,16,17,21] on 325 extracted teeth, and three are in vitro studies [18,19,20] on 115 artificial teeth.

### 3.3. In Vivo Studies

Three [9,12,14] of the six in vivo studies used sodium hypochlorite as an irrigant, two [11,13] used a photosensitiser, and one [10] used a mixture of sodium hypochlorite and radiopaque contrast. The percentage of Hypochlorite ranged from 2.5% to 6%.

The six in vivo studies applied different irrigant activation techniques. One [9] study utilised negative apical pressure, and one [10] used an Erbium chromium: yttrium–scandium–gallium–garnet (Er, Cr: YSGG) laser and passive ultrasonic irrigation (PUI). Two [11,13] studies used photoactivated disinfection (PAD), one [12] used passive ultrasonic irrigation, and one [14] used ultrasonic activation.

The follow-up period varied from 6 weeks to 72 months. With each activation protocol performed, the absence of clinical pathological signs, radiographic healing of the lesion, root elongation, and apex closure were observed.

Table 1 summarises the authors, the journal, the year of publication, the type of study (in vivo), the type of irrigant used, the irrigant concentration, the type of activation used, the number and type of teeth on which the study was performed, whether or not there was extrusion, the follow-up period, and clinical and radiographic evaluations.

### 3.4. Ex Vivo and In Vitro Studies

Of the seven [15,16,17,18,19,20,21] non-in vivo studies, four [15,16,17,21] were ex-vivo and three [18,19,20] in vitro. One [15] study used a contrast solution, five [16,17,18,20,21] employed sodium hypochlorite (NaOCl), and one [19] used distilled water. The percentage of hypochlorite ranged from 1% to 6%. The flow rate of the irrigant varied between 0.25 and 0.05 mL/s. The position of the instrument tip varied from working length to working length minus 10 mm.

Four [15,16,17,21] studies used real teeth, two [18,19] used teeth 3D printed from a real tooth, and one [20] study used an artificial root. The total number of teeth was 400.

Three [15,17,19] studies measured the volume of extruded irrigant beyond the apex. One study [20] evaluated the colour change of bovine pulp tissue stained with fuchsin. Another [21] studied the discolouration of saline agar stained with 1% acid red. One study [16] evaluated discolouration in a 0.2% agarose gel containing 1 mL of 0.1% m-cresol purple. One study [18] assessed the pressure generated over the apex.

Six studies [15,16,17,19,20,21] showed extrusion beyond the apex following irrigant activation. A study [12] demonstrated that using sonic activation with EDDY at working length and the RinsEndo device caused the pressure set to be a safe limit to be exceeded in one hundred percent of cases.

A study [20] showed that the CAB technique can avoid extrusion beyond the apex of the irrigant activated with internal heating and ultrasound.

Table 2 summarizes the authors, the journal, the year of publication, the type of study (ex vivo), the type of irrigant used, the concentration, the flow rate, the total amount of irrigant, the type of activation used, the settings of instrument, the insert position, the activation protocol, the number and type of teeth on which the study was performed, whether or not there was extrusion, how much extruded, and how the extrusion was evaluated.

Table 3 summarizes the authors, the journal, the year of publication, the type of study (in vitro), the type of irrigant used, the concentration, the flow rate, the total amount of irrigant, the type of activation used, the settings of instrument, the insert position, the activation protocol, the number and type of teeth on which the study was performed, whether or not there was extrusion, how much extruded, and how the extrusion was evaluated.

## 4. Discussion

Based on the analysis of the results of the current review, the discussion was established by dividing it into two parts: the first regarding the in vivo studies and the second concerning the ex vivo and the in vitro studies. The division analysed the results better since the in vivo studies did not assess whether the irrigant was extruded beyond the apex except for one. On the other hand, the in vitro and ex vivo studies quantified the extrusion of the irrigant. These findings have significant implications for endodontics, particularly regarding the potential risks and benefits of different activation techniques.

Disinfection of the endodontic is essential to achieving long-term success of endodontic treatment and preventing failure due to the persistence of bacteria [1]. The chemical disinfection procedure is performed in a standard endodontic treatment and other procedures, such as pulpal regeneration and intentional transplantation [2,3].

### 4.1. In Vivo Studies

According to Symerman and Nosrat, activation of 6% NaOCl with Endovac Macrocannula leads to healing of the periapical lesion, partial or complete root mineralisation, and varying degrees of apex closure. However, the study was conducted on five teeth, including one lower second molar. The maximum follow-up period was 72 months [9].

Two studies used photo-activated disinfection on three single-rooted teeth and achieved healing of the periapical lesion, root elongation, and apical closure; the maximum follow-up period was 12 months [11,13].

Maniglia-Ferreira et al. performed passive ultrasonic activation of 2.5 percent sodium hypochlorite and obtained progression in root development at one and three years, assessed by CBCT [12].

According to McCabe P., the ultrasonic activation of 5% NaOCl leads to the progressive thickening of root canal walls, increased root length, and apex closure from controls up to 18 months [14].

In agreement with Peeters et al., activation with a (Er; Cr; YSGG) laser succeeds in bringing NaOCl to the apical level better than passive ultrasonic activation (PUI). Both activation techniques did not cause extrusion beyond the apex of the irrigant, assessed radiographically by mixing 2.5% NaOCl with a radiopaque contrast [10].

The in vivo studies in this review, which utilised various activation techniques, all demonstrated success in the absence of clinical pathological signs. Furthermore, they showed increased root length, thickness, and apical closure, supporting their effectiveness in endodontic treatment.

### 4.2. Ex Vivo and In Vitro Studies

This section presents the findings of studies conducted outside the living organism, providing valuable insights into the effects of different activation techniques on irrigant extrusion.

Due to its advantages, sodium hypochlorite is the main irrigant used in endodontics [22,23]. Nevertheless, it has a cytotoxic effect on the body’s tissues [24]; hypochlorite damage results from extrusion beyond the apex of the hypochlorite [2,25].

Given the serious damage that hypochlorite extrusion beyond the apex can cause, it is crucial to minimise the risk. Once irrigant extrusion occurs, immediate action is necessary, including aspirating the irrigant, rinsing the endodontic space, leaving the tooth open, and prescribing appropriate medications.

The first step in the disinfection of the endodontic space is the insertion of the irrigant through an endodontic needle. This review and other studies have shown that using a needle with a side opening is recommended to decrease the risk of extrusion of the irrigant [5,6,25,26,27,28,29,30,31]. The flow rate of the irrigant is important for the risk of extrusion. The flow rate value in the studies in this review is between 0.05 and 0.25 mL/s. The literature agrees that the lower the flow rate, the lower the risk of extrusion [29,32]. It is important to consider the various factors influencing the flow rate. Being of the male gender and developing a greater force on the syringe is associated with a higher flow rate; syringe design is also crucial, as larger syringes convey more irrigant with little plunger movement [29,32,33]. Another factor to consider is the depth at which the needle tip is placed. The closer this is to the working length, the greater the pressure generated at the apex and, thus, the greater the risk of extrusion; the results agree with other studies in the literature [18,34,35].

According to Magni et al. [18], sonic activation with EDDY leads to the development of a pressure greater than 5.73 mm Hg at the apical level and the pressure is influenced by the distance from the apex at the working length (the point at which the file reaches the end of the root canal), the pressure exceeded the threshold value in 100 percent of the cases, moving away from the working length the percentage decreased [18]. Activation with RinsEndo also caused the threshold value to be exceeded in 100% of cases and independently of the working length. However, this result does not mean that other methods cannot cause an extrusion. It must be considered that the resistance of apical tissues is not always the same. There are various factors to be considered that may decrease the resistance of the tissues to the irrigant.

For instance, the female sex is more prone to hypochlorite accidents due to greater bone laxity [36]. Younger patients have a higher risk, as root development may often need to be completed and the apex open [37]. A larger diameter increases the risk of extrusion [37]. In all the studies in this review, models with an open apex were used, and in each case, extrusion beyond the apex occurred to varying degrees, depending on the techniques used and the study conditions [15,16,17,18,19,20,21]. Older patients may have roots closer to the max mineralisation and thus less resistance to extrusion [38,39]. Bone mineralisation affects resistance so that the cortex opposes greater resistance than the spongy bone, and the maxillary bone is more subject to the risk of extrusion than the mandibular bone [24,25].

Furthermore, a patient’s position in the chair can influence venous pressure and, therefore, the resistance that the hypochlorite encounters beyond the apex [25]; the force of gravity also plays an important role, as demonstrated by the study by Sharma et al. [19].

Systemic diseases, such as osteoporosis, which cause less calcification of bone tissue, may lead to greater sequelae when NaOCl is extruded [36]. Local pathology may also affect the risk of extrusion: extrusion occurs more in teeth with necrotic pulp than in vital teeth due to the barrier effect the pulp exerts [29,36,40]. The presence of periapical pathology reduces the risk of extrusion because of the increased pressure it generates at the root apex [40].

However, when it comes to Sonic activation with EDDY, it resulted in significantly greater extrusion of irrigant over apex than positive pressure without agitation, ultrasonic agitation with Irrisonic, ultrasonic agitation with Irrisonic Power, mechanical agitation with Easy Clean, and mechanical agitation with XP-Endo Finisher [15]. The sonically activated group with EDDY showed an extrusion of 0.017 mL with a maximum value of 0.11 mL; all other groups produced maximum extrusions of less than 0.1 mL [15]. According to other studies, 0.1 mL or more of sodium hypochlorite over the apex reduces cell viability by 40% [41,42].

The working method of the sonic instrument can justify the higher extrusion because it has a wide oscillatory motion that promotes acoustic streaming and cavitation [43]. In addition, the sonic instrument also has a larger diameter, making it difficult for the irrigant to flow out coronally [15].

Other studies have shown that all groups showed some degree of extrusion, which applies to mature teeth with closed apices [44].

One study compared needle irrigation, ultrasonic activation, and sonic activation with EDDY, an Erbium: yttrium–aluminum–garnet (Er: YAG) laser, and activation with a diode laser [16]. The results are similar to those of dos Reis’ study. Activation with EDDY results in greater extrusion of irrigant beyond the apex than the other groups [16]. No statistically significant difference exists between the EDDY and Er: YAG laser and needle groups [16].

The erbium laser’s short activation time may explain its lack of difference from the other groups [16].

One reason the erbium laser and the diode laser differ in their different modes of use is that the former is used in pulse mode, and the latter is used in continuous mode [16]. Each pulse of the laser causes an acceleration in the irrigant [45] strong enough to make it extrude past the apex [46].

Velmurugan et al. [17] compared extrusion of irrigant (3% NaOCl) over apex comparing Endovac Microcannula, Endovac Microcannula, NaviTip irrigation needle and Max-i-Probe irrigation needle. All groups showed extrusion beyond the apex [17]. The NaviTip and Max-i-Probe groups extruded all 9 mL of hypochlorite used, the group with Endovac Microcannula extruded about 7.5 mL out of 9 mL, and the Endovac Microcannula group extruded about 0.20 mL out of 9 mL used [17]. The results of this study are similar to those of Mitchell et al. that, even in mature teeth, Endovac extrudes in fewer specimens [47].

According to the authors, the lower extrusion of the Endovac units is due to both instruments aspirating the irrigant. The marked difference between the Endovac microcannula and microcannula may lie in the different instrument diameters (0.55 mm for the microcannula and 0.10 mm for the microcannula) [17].

The NaviTip (closed-ended) and Max-i-Probe (open-ended) groups recorded the same result, in contrast to the study by Psimma et al., in which the side-opening needle caused less extrusion of irrigant beyond the apex than the apex-opening needle [30].

In agreement with previous papers by dos Reis et al. [15] and Karasu et al. [16], Magni et al. [18] have also proven that sonic activation with EDDY carries an increased risk of extrusion beyond the apex. In particular, the greater the risk, the closer the sonic tip approaches the apex.

The RinsEndo group recorded the highest apical pressure values at all insertion depths [18]. Pressure values that exceeded the established limit were also recorded in the LiteTouch-Induced Photomechanical Irrigation (LT-IPI) group with settings at 20 mJ and 50 Hz [18].

Activation with EDDY and RinsEndo should be used cautiously because it is more likely to carry the irrigant past the apex in teeth with an open apex [18].

Ultrasonically activated irrigation (UAI), EndoVac, Self-Adjusting file (SAF), and XP-endo finisher did not generate pressure values above the threshold, regardless of instrument settings and insertion depth [18].

This finding does not mean that other activation techniques cannot cause extrusion. Conditions such as maxillary location and being of the female sex can lead to lower periapical pressure [47] and continuity of the apices with the sinus [18].

The study by Jamleh et al. [21] disagrees with Sharma et al. [19] regarding using EndoVac and positive pressure.

According to Jamleh et al. [21], activation with EndoVac induces greater extrusion beyond the apex of the irrigant than positive pressure; Sharma et al. [19] demonstrated the opposite.

The difference may lie in the fact that Sharma et al. used only the macrocannula, while Jamleh et al. [21] used macro- and microcannula in sequence.

Looking at the other studies, we note that when EndoVac was used, the authors always separated the use of the macrocannula from the microcannula [17,18]. In fact, when micro- and macro cannula were investigated separately, the greatest extrusion beyond the apex occurred in the groups with the microcannula [17,18].

Using EndoVac microcannula and EndoVac macrocannula sequentially in the same group may have made the results compatible with those of the EndoVac microcannula groups in the other studies.

The application of a plug at the apical level prevents hypochlorite leakage, even after activation [20]. Placing a collagen plug in saline solution (CAB technique) was verified to be effective in preventing hypochlorite leakage and, additionally, in vivo application has resulted in wound healing [20].

## 5. Limitations and Future Developments

This study is the first on this topic.

As weaknesses, the various in vivo studies use different activation techniques with different irrigants or, when equal, with different concentrations. The patient sample is limited, and they were all performed on single-rooted teeth; only one study successfully applied the protocol to a multi-rooted analysis.

Future studies could focus on analysing irrigant extrusion in vivo, which currently needs to be sufficiently investigated.

It would be interesting to understand whether it is possible to perform apexification after disinfection without irrigant activation and then, after apical stop creation, perform endodontic treatment with irrigant activation. Having created a stop in the apex will result in less chance to extrude.

For both the in vitro and ex vivo studies, the analysed studies did not use the same tooth models, but the sample analysed was bigger than the in vivo. As with in vivo studies, the type of irrigant and its concentrations vary from study to study, as does the method used to measure irrigant extrusion beyond the apex.

Due to the design of the in vitro/ex vivo studies, most of the subject-related variables should have been considered, which makes this difficult. Three studies measure the volume of extruded irrigant beyond the apex; beyond the apex, there is nothing to resist extrusion, which is not the case in vivo [10,12,14]. In three other studies beyond the apex, there is agarose gel or pulp, which, to some extent, generates resistance to the irrigant’s extrusion and periodontal tissues [analysed].

Further future studies on the topic may be research with more scientific evidence such as: systematic reviews, meta-analyses or umbrella reviews.

## 6. Conclusions

Based on the studies analysed in this review, all of the activation techniques investigated can cause extrusion beyond the apex of the irrigant. Extrusion may not always be clinically relevant. However, the consequences of excessive irrigant leakage from the apex are dangerous, so we have tried to assess all the variables that may cause it (needle without lateral opening; high flowrate; depth at which the needle is placed; techniques such as sonic activation with EDDY and activation with RinsEndo; lower apical tissue resistance due to: presence of sinuses, tissue laxity, bone hypomineralisation, female gender, young age, degree of apex maturation; and a patient’s position in the chair) and adopt techniques to reduce it.

Some activation techniques, such as EDDY sonic activation, can increase irrigant extrusion. Increased hypochlorite activation by these techniques may also increase the cytotoxic effect on apical cells. Considering that the apex is open and therefore root development is not yet finished using techniques such as sonic activation can lead to greater tissue damage.

Techniques such as the CAB technique can prevent the irrigant from extruding from the immature apex.

## Figures and Tables

**Figure 1 jcm-13-06611-f001:**
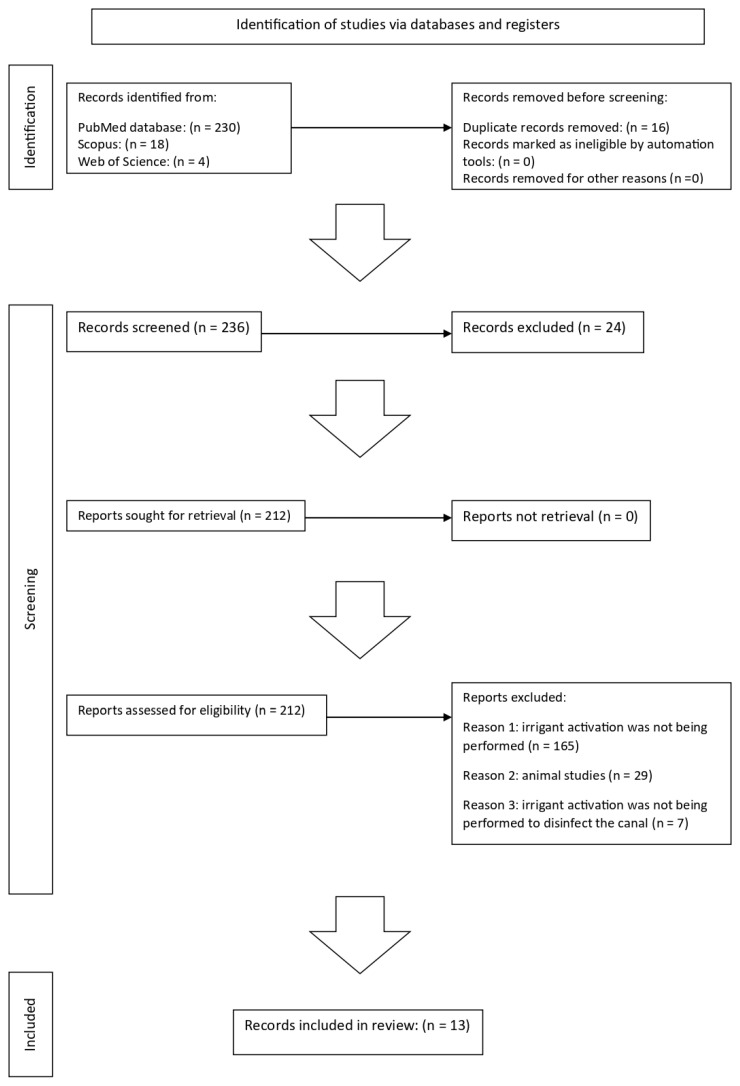
Study selection flowchart.

**Table 1 jcm-13-06611-t001:** Results of in vivo studies.

Studies	Type of Study	Irrigant		Activation	No. of Teeth	Type of Teeth	Extrusion Over Apex	Follow-up	Clinical Evaluation	Radiographic Evaluation
Cymerman and Nosrat. J. Endod. 2019 [9]	In vivo	6% NaOCl	17% EDTA	EndoVac Macrocannula	5	11 (2), 12, 21, 47	N.A.	From 20 to 72 months	No pain. Probing depths in normal limits. Mobility in normal limits.	Complete osseous healing of the periapical lesions. The root canal spaces showed partial to complete mineralization. Varying degrees of apical closure.
Peeters et al. Iran. Endod. J. 2018 [10]	In vivo	Mixture of radiopaque contrast medium and 2.5% NaOCl Ratio of 40:60		(Er,Cr:YSGG) laser	20 of 40	Teeth with single root canals	No	N.A.	No pain. No swelling. Probing depths in normal limits.	The contrast had reached the apex successfully in 20 teeth.
				PUI	20 of 40	Teeth with single root canals	No			The contrast had reached the apex successfully in 9 teeth; in 11 teeth it had not reached apex.
Abdel Hafiz Abdel Rahim et al. F1000Res. 2019 [11]	In vivo	Photosensitizer solution		PAD	1	11	N.A.	6, 9, 12 months	No adverse signs and symptoms.	Increase in root length and root thickness at six months and complete root closure at 12 months.
Maniglia-Ferreira et al. J. Endod. 2017 [12]	In vivo	2.5% NaOCl	17% EDTA	PUI	1	11	N.A.	1, 3, 6, 12, 36 months	No pain. No swelling. No fistula.	CBCT scans obtained at the 1 year and 3-year follow-up revealed progression of root development
Johns et al. J. Conserv. Dent. 2014 [13]	In vivo	Photosensitizer solution		PAD	2	11, 21	N.A.	6, 10 months	Asymptomatic	Continued thickening of the dentinal walls, root lengthening, regression of the peri-apical lesion and apical closure.
McCabe P. Int. Endod. J. 2015 [14]	In vivo	5% NaOCl		Ultrasonic activation	1	21	N.A.	6 weeks, 3, 6, 12, 18 months	Asymptomatic	Progressive thickening of the root canal walls, root lengthening and apical closure.

NaOCl = sodium hypochlorite; EDTA = ethylenediaminetetraacetic acid; N.A. = not applicable; Er,Cr:YSGG = Erbium chromium: yttrium–scandium–gallium–garnet; PAD = photo-activated disinfection; PUI = passive ultrasonic irrigation; EndoVac = negative pressure irrigation.

**Table 2 jcm-13-06611-t002:** Results of ex vivo studies.

Studies	Type of Study	Irrigant Concentration Flow Rate Total Amount			Activation		Settings	Insert Position	Activation Protocol	No. of Teeth	Type of Teeth	Extrusion Over Apex	Quantity of Extruded Irrigant	Extrusion Evaluation Method
dos Reis et al. J. Endod. 2020 [15]	Ex vivo	Contrast solution: Ioditrast 76 N.A. 0.25 mL/s 15 mL			PP and no agitation		M.D.	W.L. −1 mm	5 mL/20 s × 3 times (irrigation)	15 of 15	Single root mandibular premolars	Yes	0.76 (0.0–17.40)	Volume of extruded irrigant in mL
					Ultrasonic agitation with Irrisonic		5 (25%)	W.L. −1 mm	20 s × 3 times	15 of 15	Single root mandibular premolars	Yes	2.28 (0.0–27.57)	
					Ultrasonic agitation with Irrisonic Power		7 (35%)	W.L. −1 mm	20 s × 3 times	15 of 15	Single root mandibular premolars	Yes	3.14 (0.0–29.23)	
					Mechanical agitation with Easy Clean		1200 rpm 1.0 Ncm	W.L. −1 mm	20 s × 3 times	15 of 15	Single root mandibular premolars	Yes	3.15 (0.0–57.93)	
					Mechanical agitation with the XP-endo Finisher		800 rpm 1.0 Ncm	W.L. −1 mm	20 s × 3 times	15 of 15	Single root mandibular premolars	Yes	0.67 (0.0–9.31)	
					Sonic agitation with EDDY		M.D.	W.L. −1 mm	20 s × 3 times	15 of 15	Single root mandibular premolars	Yes	17.19 (0.0–110.40)	
Karasu et al. Lasers Med. Sci. 2022 [16]	Ex vivo	NaOCl M.D. 0.1 mL/s 6 mL			Needle irrigation		N.A.	W.L. −2 mm	6 mL/60 s (irrigation)	15 of 75	Maxillary anterior teeth with single root	Yes	1.57 (1.45 ± 0.38)	Pixel percentages discoloration in a 0.2% agarose gel containing 1 mL of 0.1% m-cresol purple.
					Ultrasonic irrigation		M.D.	W.L. −2 mm	20 s × 3 times	15 of 75	Maxillary anterior teeth with single root	Yes	0.81 (0.80 ± 0.22)	
					Sonic agitation with EDDY		M.D.	W.L. −2 mm	20 s × 3 times	15 of 75	Maxillary anterior teeth with single root	Yes	2.53 (2.28 ± 0.66)	
					Er:YAG laser		1 W, 12 Hz, 85 mJ	W.L. −3 mm	5 s × 5 times (10 s interval)	15 of 75	Maxillary anterior teeth with single root	Yes	2.06 (1.86 ± 0.71)	
					Diode laser		1 W	W.L. −2 mm	6 s × 4 times (10 s interval)	15 of 75	Maxillary anterior teeth with single root	Yes	0.71 (0.67 ± 0.40)	
Velmurugan et al. J. Dent. (Tehran). 2014 [17]	Ex vivo	NaOCl 3% 0.15 mL/s 9 mL			EndoVac Microcannula		N.A.	W.L. −1 mm	60 s (irrigation)	20 of 80	Maxillary central incisors	Yes (100%)	7.53 mL of 9 mL used	Volume of extruded irrigant in mL
					EndoVac Macrocannula		N.A.	W.L. −1 mm	60 s (irrigation)	20 of 80	Maxillary central incisors	Yes (40%, 8/20)	0.23 mL of 9 mL used	
					NaviTip irrigation needle		29 gauge	W.L. −2 mm	60 s (irrigation)	20 of 80	Maxillary central incisors	Yes (100%)	9 mL of 9 mL used	
					Max-i-Probe Irrigating needle		M.D.	W.L. −2 mm	60 s (irrigation)	20 of 80	Maxillary central incisors	Yes (100%)	9 mL of 9 mL used	
Jamleh et al. Clin. Oral. Investig. 2016 [21]	Ex vivo	NaOCl 6% 0.05 mL/s 6 mL	EDTA 17% 0.05 mL/s 9 mL	NaOCl 6% 0.05 mL/s 6 mL	iNP system		−20 kPa	W.L. −2 mm	N.A.	10	Single canalled lower incisors	Yes	2/10	Discoloration of warm normal saline agar coloured with 1% acid red
		NaOCl 6% 0.05 mL/s 2 mL	NaOCl 6% 0.05 mL/s 2 mL	EDTA 17% 0.05 mL/s 2 mL	NaOCl 6% 0.05 mL/s 2 mL	EndoVac microcannula + microcannula	−20 kPa	W.L.	N.A.	10		Yes	10/10	
		NaOCl 6% 0.05 mL/s6 mL	EDTA 17% 0.05 mL/s 9 mL	NaOCl 6% 0.05 mL/s 6 mL	PP		−20 kPa	W.L. −2 mm	N.A.	10		Yes	9/10	
		Distilled water N.A. 0.05 mL/s 21 mL			PP (Control)		M.D.	M.D.	M.D.	10		N.A.	N.A.	

N.A. = not applicable; M.D. missing data; NaOCl = sodium hypochlorite; W.L. = working length; Er,Cr:YSGG = Erbium chromium: yttrium–scandium–gallium–garnet; PUI = passive ultrasonic irrigation; PP = positive pressure; EndoVac = negative pressure irrigation; Er:YAG = Erbium: yttrium–aluminum–garnet; UAI = Ultrasonically activated irrigation; SAF = Self-adjusting file; LT-IPI = LiteTouch-Induced Photomechanical Irrigation; RisEndo = hydrodynamic activation; IH = internal heating; iNP = Intracanal negative pressure needle.

**Table 3 jcm-13-06611-t003:** Results of in vitro studies.

Studies	Type of study	Irrigant Concentration Flow Rate Total Amount	Activation	Settings		Insert Position	Activation Protocol	No. of Teeth	Type of Teeth	Extrusion Over Apex	Quantity of Extruded Irrigant	Extrusion Evaluation Method
Magni et al. Int. Endod. J. 2021 [18]	In vitro	Na1 mL M.D.0.1 mL + 2.8 mL additional for EndoVac	UAI	10 (intensity)		W.L.	10 s × 10 times	1	3D-printed central maxillary incisor	NA	0	Percentage > 5.73 mm Hg (%)
						W.L. −1 mm		1			0	
						W.L. −2 mm		1			0	
						W.L. −3 mm		1			0	
				20 (intensity)		W.L.		1			0	
						W.L. −1 mm		1			0	
						W.L. −2 mm		1			0	
						W.L. −3 mm		1			0	
				30 (intensity)		W.L.		1			0	
						W.L. −1 mm		1			0	
						W.L. −2 mm		1			0	
						W.L. −3 mm		1			0	
			Sonic agitation with EDDY	M.D.		W.L.	10 s × 10 times	1	3D-printed central maxillary incisor	NA	100	
						W.L. −1 mm		1			30	
						W.L. −2 mm		1			10	
						W.L. −3 mm		1			20	
			EndoVac	Macrocannula	N.A.	W.L.	10 s × 10 times	1	3D-printed central maxillary incisor	NA	0	
						W.L. −1 mm		1			0	
						W.L. −2 mm		1			0	
						W.L. −3 mm		1			0	
				Microcannula	N.A.	W.L.	10 s × 10 times	1			0	
						W.L. −1 mm		1			0	
						W.L. −2 mm		1			0	
						W.L. −3 mm		1			0	
			SAF	M.D.		W.L.	10 s × 10 times	1	3D-printed central maxillary incisor	NA	0	
						W.L. −1 mm		1			0	
						W.L. −2 mm		1			0	
						W.L. −3 mm		1			0	
			Mechanical agitation with the XP-endo Finisher	1000 rpm		W.L.	10 s × 10 times	1	3D-printed central maxillary incisor	NA	0	
						W.L. −1 mm		1			0	
						W.L. −2 mm		1			0	
						W.L. −3 mm		1			0	
			LT-IPI	20 mJ/10 Hz		W.L. −6 mm	10 s × 10 times	1	3D-printed central maxillary incisor	NA	0	
						W.L. −10 mm		1			0	
				20 mJ/20 Hz		W.L. −6 mm		1			0	
						W.L. −10 mm		1			0	
				20 mJ/50 Hz		W.L. −6 mm		1			50	
						W.L. −10 mm		1			10	
				40 mJ/10 Hz		W.L. −6 mm		1			0	
						W.L. −10 mm		1			0	
			RinsEndo	1.6 Hz, 105 µL/s		W.L.	10 s × 10 times	1	3D-printed central maxillary incisor	NA	100	
						W.L. −3 mm		1			100	
						W.L. −4.5 mm		1			100	
						W.L. −6 mm		1			100	
						W.L. −10 mm		1			100	
Sharma et al. Eur. Endod. J. 2020 [19]	In vitro	Distilled water N.A. 0.062 mL/s ^1^ 0.066 mL/s ^2^ 0.083 mL/s ^3^ 10 mL	PP ^1^	Maxilla	M.D.	W.L. −2 mm	M.D.	5 of 30	3D maxillary central incisor	Yes	0.6 (0.5–1.05) mL	Volume of extruded irrigant in mL
				Mandible				5 of 30		Yes	10 (10–10) mL	
			PUI ^2^	Maxilla	M.D.	W.L. −2 mm	20 s × 5 times	5 of 30		Yes	1 (0.5–1.50) mL	
				Mandible				5 of 30		Yes	10 (10–10) mL	
			EndoVac ^3^	Maxilla	M.D.	W.L. −2 mm	M.D.	5 of 30		Yes	0 (0–0.2) mL	
				Mandible				5 of 30		Yes	0.5 (0.5–1.5) mL	
Iandolo et al. J. Conserv. Dent. 2021 [20]	In vitro	NaOCl 5.25% M.D. 6 mL	IH + ultrasonic activation (without CAB technique)	180 °C + M.D.		W.L. −3 mm	(8 s + 30 s) × 5 times	20	Artificial root canal	Yes	100%	Discoloration of fuchsine-stained bovine pulp tissue
			IH + ultrasonic activation (with CAB technique)	180 °C + M.D.		W.L. −3 mm	(8 s + 30 s) × 5 times	20		No	0%	

M.D. = missing data; NaOCl = sodium hypochlorite; W.L. = working length; SAF = Self-adjusting file; N.A. = not applicable; PUI = passive ultrasonic irrigation; PP = positive pressure; EndoVac = negative pressure irrigation; UAI = Ultrasonically activated irrigation; LT-IPI = LiteTouch-Induced Photomechanical Irrigation; RisEndo = hydrodynamic activation; IH = internal heating. ^1,2,3^ The flow rate is different depending on the techniques used. For PP activation the flow rate is 0.062 mL/s; for PUI it is 0.066 mL/s; for EndoVac it is 0.083 mL/s.

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
