# Peer review of "Activation of Irrigants in Root Canals with Open Apices: A Narrative Review"

_jcm, 2024, doi:10.3390/jcm13216611_

Round 1

Reviewer 1 Report

Comments and Suggestions for Authors

Dear authors,

The review entitled “Activation of irrigants in root canals with open apices. A narrative review” is extremely welcome to complete the literature data and to improve the knowledge of professionals in the field of dental practice with significant implications for endodontics, particularly regarding the potential risks and benefits of different activation techniques.

I have some remarks for the authors:

·         In Introduction it would be useful to mention the irrigants that can be used to disinfect the canal

·         In 2. Materials and Methods….the authors mentioned: “The authors meticulously collected scientific articles on activating irrigants in tooth roots with open apex until 15 September 2023….”

It is necessary to use:  Were meticulously collected scientific articles on activating irrigants in tooth roots with open apex….Also, it is necessary to mention the time interval during which the data were searched.

·         In Results, at page 3 must be specified the title of the flow chart !

·         In Table 1. Results of in vivo studies…..Do not use N.of teeth….to be used No. of teeth

·         Exactly the same remark for Table 2…..Do not use N.of teeth….to be used No. of teeth

·         Table 2. Results of ex vivo and in vitro studies…………….it is necessary to reconsider this table!

 I consider that it is too long in this present form. For example: the authors can consider in one table Results of ex vivo studies and in another table Results of in vitro studies

·         At line 211, the authors mentioned: “Systemic pathologies may lead to major sequelae in the case of NaOCl extrusion”…..It would be useful to mention the systemic pathologies !

·         In 5. Conclusions

The authors mentioned: “Based on the studies analysed in this review, all of the activation techniques investigated can, to varying degrees, cause extrusion beyond the apex of the irrigant”……do not use “to varying degrees”!

The authors mentioned: “However, the consequences of excessive irrigant leakage from the apex are dangerous, so try to assess all the variables that may cause it (Here should be briefly mentioned the variables that may cause it)  and adopt techniques to reduce it.

It is necessary as conclusion to highlight the potential risks and benefits of different activation techniques.

Author Response

We really appreciate the time and work you invested in reviewing our manuscript and helping us improve the quality of our work. We have taken your suggestions on board and edited the manuscript.

Attached you will find the answer to the review
thank you very much

Reviewer 2 Report

Comments and Suggestions for Authors

The topic is interesting and useful and has clinical application.

ABSTRACT

The abstract summarizes the content of the work in a clear and objective manner.

INTRODUCTION

The introduction is too short. What irrigators? What activation?...What irrigation techniques are there?

MATERIALS AND METHODS

Why didn't you search on Embase?

Why didn't they use research equations for each base as is usually and correctly done?

“… articles in English…”  - It is not part of the inclusion/exclusion criteria - it is a filter.

“…No time filters were applied to the search…” - But were there other filters applied?

The authors followed the methodology of an SR because they did not construct a PICO question?

RESULTS

They are well systematized and presented. They correspond to what was obtained once the methodology chosen by the authors and which is used in the SR was applied.

The figures and tables are correct and well presented, but the caption for the figure on page 3 is missing.

DISCUSSION

“…into two paragraphs…” – Line 2. correct: “…into two parts…”.

“…Disinfection of the endodontic is essential to achieving long-term success of endodontic treatment and not failure due to bacteria persistence [17]. The chemical disinfection procedure is performed in standard endodontic treatment and other procedures such as pulpal regeneration and intentional transplantation [18,19]…” - This is the content of the introduction.

Limitations and Future Developments

The authors do not consider doing: SR; Meta-analysis; Umbrella Review;...

CONCLUSIONS

They are short, objective and clear, as they should be.

Author Response

(The authors gave the same response as above.)

Round 2

Reviewer 1 Report

Comments and Suggestions for Authors

I consider that the article can be published in this latest version.

Author Response

(The authors gave the same response as above.)

Reviewer 2 Report

Comments and Suggestions for Authors

Comment of the Reviewer: Why didn't they use research equations for each base as is usually and
correctly done?
Author’s reply: We are sorry if our explanation of Materials and Methods was not precise. We have
used the same keywords ("Open Apex" OR "Immature Apex" OR "Immature Teeth" AND
"Irrigation Activation" OR "Activated Irrigation" OR "Irrigation OR Activation”) for all three
databases.

- The thing is that the search equations are not the same for all databases.

Author Response

(The authors gave the same response as above.)
